# A fungal pathogen deploys a small silencing RNA that attenuates mosquito immunity and facilitates infection

Chunlai Cui [1,2,4], Yan Wang [1,2,4], Jingnan Liu [3], Jing Zhao[1], Peilu Sun [1] & Sibao Wang [1]

Insecticidal fungi represent a promising alternative to chemical pesticides for disease vector control. Here, we show that the pathogenic fungus *Beauveria bassiana* exports a microRNA-like RNA (bba-milR1) that hijacks the host RNA-interference machinery in mosquito cells by binding to Argonaute 1 (AGO1). bba-milR1 is highly expressed during fungal penetration of the mosquito integument, and suppresses host immunity by silencing expression of the mosquito Toll receptor ligand Spätzle 4 (Spz4). Later, upon entering the hemocoel, bba-milR1 expression is decreased, which avoids induction of the host proteinase CLIPB9 that activates the melanization response. Thus, our results indicate that the pathogen deploys a cross-kingdom small-RNA effector that attenuates host immunity and facilitates infection.

[1] CAS Key Laboratory of Insect Developmental and Evolutionary Biology, CAS Center for Excellence in Molecular Plant Sciences, Institute of Plant Physiology and Ecology, Shanghai Institutes for Biological Sciences, Chinese Academy of Sciences, Shanghai 200032, China. [2] University of Chinese Academy of Sciences, Beijing 100049, China. [3] School of Life Science and Technology, ShanghaiTech University, Shanghai 201210, China. [4] These authors contributed equally: Chunlai Cui, Yan Wang. Correspondence and requests for materials should be addressed to S.W. (email: sbwang@sibs.ac.cn)

Malaria remains one of the world's deadliest infectious diseases, with 435,000 global deaths and 219 million new cases in 2017, up from 217 million cases in 2016[1]. Because mosquitoes are obligatory vectors for malaria transmission[2], vector control represents the front-line intervention tool for preventing malaria transmission[3]. However, vector control efforts are being undermined by increasing insecticide resistance[4], and alternative ecologically sound approaches to control mosquitoes are urgently needed. The insect pathogenic fungus *Beauveria bassiana* (Cordycipitaceae) offers a promising environment-friendly alternative to chemical insecticides[5,6]. This fungus is effective at killing both insecticide-resistant and -susceptible adult mosquitoes[7,8] and considered as a next-generation mosquito control agent[9,10]. However, fungal pathogens kill mosquitoes relatively slow compared with chemical pesticides, which has hampered their widespread application[11]. Thus, better understanding of molecular interactions between the fungus and the host mosquito is of particular importance for improvement of its efficacy.

Insects employ innate immune responses to fight pathogenic infection. Toll is the principal antifungal pathway by directing the production of antifungal peptides[12]. Melanization is another important facet of the mosquito immune defense against fungi[13]. To cause infection, pathogens must counteract their host immune responses[14]. In case of fungi, the molecular mechanism of host immunity suppression has not been described, although the metabolites secreted by insect fungal pathogens have been implicated in this phenomenon[15–17]. Plant pathogenic fungi are known to deliver cell-entering effectors to suppress host immunity[18,19]. However, to our knowledge, no examples of such effector molecules have been described for insect pathogenic fungi[20].

Small silencing RNAs (sRNAs) are short noncoding RNAs that regulate gene expression by binding to Argonaute (AGO) proteins and directing the RNA-induced silencing complex (RISC) to RNAs with complementary sequences. miRNAs are a class of sRNAs of ~22 nt, and are important modulators of various biological processes, including development and host–microorganism interactions[19,21,22].

Here, we report that the insect fungal pathogen *B. bassiana* produces a microRNA-like sRNA bba-milR1 that acts as a host cell-entering effector to modulate the expression of mosquito immune genes, demonstrating the fungal pathogen has adapted the cross-kingdom RNA interference-mediated defense mechanism during the evolutionary arms race with the insect hosts.

## Results

**Beauveria bassiana expresses miRNA-like small RNAs.** To explore possible roles of miRNAs in mosquito–fungus interactions, we profiled sRNA libraries generated from *Beuaveria bassiana*-infected *Anopheles stephensi* mosquitoes collected at 36, 60, and 84 h post fungal topical inoculation. We identified four miRNA-like small RNAs (milRNAs) whose sequences cannot be mapped to the *A. stephensi* genome, but perfectly matched the *B. bassiana* genome (Supplementary Fig. 1a). These 4 milRNAs (bba-milR1, bba-milR2, bba-milR3, bba-milR4) had predicted miRNA-like precursor stem-loop structures (Supplementary Fig. 1b), suggesting that *B. bassiana* may generate miRNA-like small RNAs during host infection. Fungi have diverse milRNA biogenesis pathways, and some milRNAs are Dicer independent[23]. *B. bassiana* has two Dicer proteins Dicer1 and Dicer2. To investigate whether *B. bassiana* milRNAs are produced by Dicer, we generated *Dicer1* deletion mutant (ΔDcl1), *Dicer2* deletion mutant (ΔDcl2), and double-mutant ΔDcl1/Dcl2 by homologous replacement (Supplementary Fig. 2a, d–f). We found that bba-milR1 could not be detected in ΔDcl2 and ΔDcl1/Dcl2

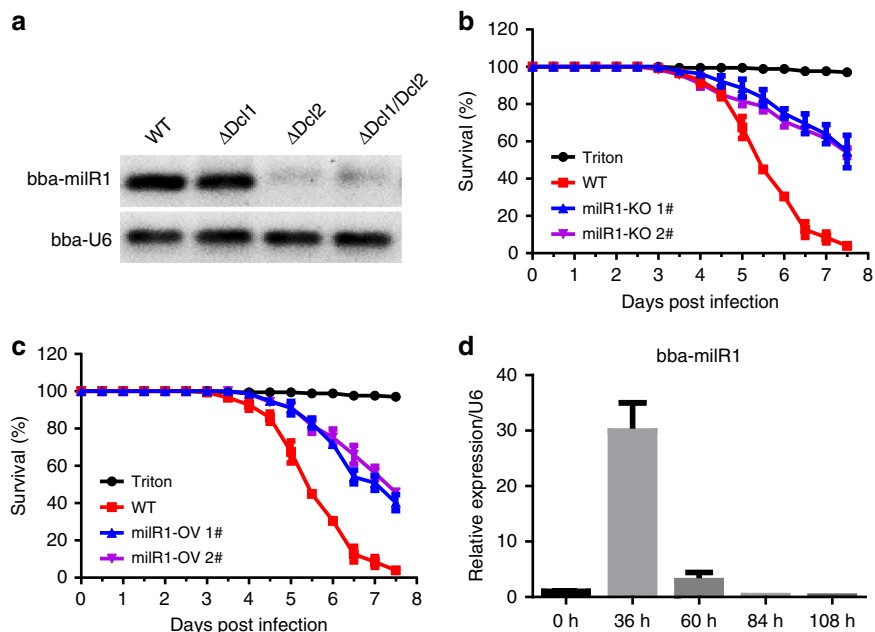

**Fig. 1** Roles of *B. bassiana* bba-milR1 on fungal pathogenicity against mosquitoes. **a** bba-milR1 cannot be detected in *B. bassiana* Dicer2 deletion mutant (ΔDcl2) and *Dicer1/Dicer2* double-mutant (ΔDcl1/Dcl2), as revealed by RT-PCR. RNA was extracted from fungal mycelium. **b** Survival of adult female *A. stephensi* mosquitoes infected with the wild-type (WT) ARSEF252 and two mutant strains milR1-KO 1# and milR1-KO 2# following topical application of a spore suspension ($10^7$ conidia/ml). Mosquitoes sprayed with sterile 0.01% Triton X-100 were used as a negative control (Triton). **c** Survival of female adult mosquitoes following topical application of a suspension of $10^7$ conidia/ml of WT and two overexpression strains milR1-OV 1# and milR1-OV 2#. The statistical significance of survival curves was analyzed with the Log-rank test. **d** Expression of bba-milR1 during *B. bassiana* ARSEF252 infecting *A. stephensi*. RNA was extracted from fungus-infected mosquitoes. The *B. bassiana* U6 small nuclear RNA (U6) was used as an internal reference in qRT-PCR assays. The expression values are normalized to time 0. The data represent three biological repeats with three technical replicates and are shown as mean ± s.e.m. Source data are provided as a Source Data file

mutants (Fig. 1a), demonstrating that bba-milR1 is Dicer dependent, whereas bba-milR2, bba-milR3, and bba-milR4 are still expressed in *Dcl1* or *Dcl2* single- and double-mutants (Supplementary Fig. 1c).

**bba-milR1 facilitates fungal infection**. To investigate whether these milRNAs are involved in fungal pathogenicity, we generated deletion mutants by separately disrupting the precursor sequences of the four milRNAs in *B. bassiana* ARSEF252 by homologous replacement (Supplementary Fig. 2a, g–j). Pathogenicity assays against *A. stephensi* showed that deletion of bba-milR1 resulted in a significant reduction in fungal virulence to mosquitoes relative to wild-type *B. bassiana* ARSEF252 ($P < 0.0001$, Log-rank test) (Fig. 1b). However, deletion of bba-milR2, bba-milR3, or bba-milR4 did not alter fungal virulence (Supplementary Fig. 3). We next generated the transgenic strains that overexpress bba-milR1, bba-milR2, bba-milR3, or bba-milR4 under control of the constitutive glyceraldehyde-3-phosphate dehydrogenase (*gpd*) promoter. For each milRNA, a fragment of ~400 bp primary milRNA centered on the genome sequence of *B. bassiana* ARSEF252 was amplified and cloned into the fungal expression vector. The overexpression vectors were separately transformed into ARSEF252 (Supplementary Fig. 2b). We validated that expression level of the four miRNAs was significantly higher in transgenic strains compared with wild-type (Supplementary Fig. 4a). These results further confirm that these milRNAs are transcribed from the genome of *B. bassiana*. Overexpression of the bba-milR2, bba-milR3, or bba-milR4 did not affect fungal virulence (Supplementary Fig. 4b), whereas overexpression of bba-milR1 (milR1-OV) resulted in significant decrease in virulence to adult *A. stephensi* mosquitoes ($P < 0.0001$, Log-rank test) (Fig. 1c) and had no obvious negative impact on fungal development (Supplementary Fig. 2c), suggesting that constitutive expression of bba-milR1 impacts fungal pathogenicity. To further investigate the function of bba-milR1 in fungus–mosquito interactions, we quantified its expression pattern during *B. bassiana* infection of *A. stephensi*. bba-milR1 was induced by ~30-fold at 36 h post infection (hpi), and then declined to very low levels as the fungus enters the host's hemocoel at about 60 hpi (Fig. 1d). To investigate whether bba-milR1 is expressed by other *B. bassiana* strains, 5-day-old female *A. stephensi* mosquitoes were infected with ARSEF252 and two other *B. bassiana* strains ARSEF2860 and Bb-bm01. We found that bba-milR1 is also detected in mosquitoes infected with these two strains (Supplementary Fig 5), suggesting that bba-milR1 is conserved among *B. bassiana* strains.

**bba-milR1 binds to mosquito Argonaute 1**. Recent studies showed that microorganism-encoded miRNAs can act as cross-kingdom regulators of host gene expression[19,21,22]. Given the high expression level of bba-milR1 at early stages of infection, we hypothesized that it may play a role in the interaction between fungus and host. To investigate whether *B. bassiana* bba-milR1 can enter mosquito cells, we incubated *Aedes albopictus* C6/36 cells with Cy3-labeled bba-milR1 and tracked RNA localization. As shown in Fig. 2a, bba-milR1 was loaded in vesicles that were present outside the cells, on the cell surface, and was transported into insect cells. These results suggest that bba-milR1 enters host cells via vesicles. Similar results were observed in *Drosophila* S2 cells (Supplementary Fig. 6). To examine the transport of milRNA in vivo, Cy3-labeled bba-milR1 was injected into the mosquito hemocoel and found that Cy3-labeled bba-milR1 was present in the cytoplasm of fat body cells (Fig. 2b).

In insects, miRNAs are preferentially assembled with Argonaute 1 (AGO1) to form an RNA-induced Silencing Complex

(RISC) that modulates gene expression[24]. To investigate whether bba-milR1 associates with RISC, we performed RNA immunoprecipitation (RIP) assays using an *A. stephensi* AGO1 (AsAGO1) antibody and assayed for AGO1-bound miRNAs using Reverse Transcription PCR (RT-PCR). As a positive control, *A. stephensi* derived miR-10-5p and miR-2940-3p were detected in AsAGO1-RIP samples (Fig. 2c). We found that *B. bassiana* milR1, milR2, milR3, and milR4 were specifically detected in AGO1-RIP samples (Fig. 2c). These results indicate that miRNAs produced by *B. bassiana* are exported into host cells and bind to mosquito AGO1 to hijack host the RNA interference (RNAi) machinery.

**bba-milR1 targets the mosquito host genes *Spz4* and *CLIPB9***. To investigate whether bba-milR1 targets mosquito genes, three stringent target prediction softwares (microTar, PITA, and miRanda) were used to predict potential targets in the 3′UTR, 5′UTR and coding sequences of the *A. stephensi* genome (Supplementary Table 1; Fig. 3a). We found that bba-milR1 could base pair with some mosquito immunity genes, including genes encoding thioester-containing protein, CLIP-domain serine protease, serine protease inhibitor, and Spätzle-like cytokine (Supplementary Table 1). To validate the interaction between bba-milR1 and the predicted immune-related target genes, we performed target verification tests ex vivo and in vivo. To verify these targets ex vivo, we cloned the DNA sequences surrounding predicted target sites using *A. stephensi* cDNA as templates into psiCheck2 vectors downstream Renilla luciferase stop codon. The engineered vectors and bba-milR1 mimics were co-transfected into HEK293T cells, and measured luciferase activity. Dual-luciferase reporter assays showed that bba-milR1 can suppress Spz4 and induce CLIPB9 expression (Fig. 3b, c). When the regions homologous to the seed sequence of bba-milR1 were mutated in the *Spz4* and *CLIPB9* reporter constructs, the luciferase activity was not affected by bba-milR1 (Fig. 3a–c). To confirm that the regulation of the two targets was indeed triggered by bba-milR1 in vivo, we injected bba-milR1 agomir, a type of chemically modified double-stranded microRNA, into adult *A. stephensi* mosquitoes, and examined the transcript level of the target genes by qRT-PCR. We found that the transcript levels of *Spz4* and *CLIPB9* were significantly suppressed and induced in agomir-treated groups, respectively (Fig. 3d, e). These results show that the host genes *Spz4* and *CLIPB9* are targeted in the coding regions and are suppressed and induced by *B. bassiana* bba-milR1, respectively. However, ex vivo and in vivo results for other predicted targets were contradictory (Supplementary Fig. 7).

**bba-milR1 manipulates host immunity to facilitate infection**. The Toll pathway in mosquitoes and other insects operates as a primary immune defense against fungal infection[12,25]. The activation of the Toll pathway requires the binding of the cytokine Spätzle (Spz) to the Toll receptor, an interaction that transduces extracellular immune signals into cells. There are six members of the Spz family in the *A. stephensi* genome, of which Spz4 has the highest expression in the integument and fat body (Supplementary Fig. 8). To confirm that *Spz4* can be targeted by bba-milR1 during infection, we further examined the expression levels of *Spz4* in mosquitoes after infection with wild-type *B. bassiana* (WT) and bba-milR1 overexpression strain (milR1-OV). Expression of *Spz4* was significantly lower in milR1-OV-infected mosquitoes than in WT-infected mosquitoes (Fig. 4a; Supplementary Fig. 9). Meanwhile, the mRNA levels of the major antifungal effector genes encoding Cecropin 1 (CEC1) and Defensin 1 (DEF1) were also reduced in milR1-OV-infected mosquitoes (Fig. 4a; Supplementary Fig. 9), indicating that bba-milR1 can suppress *Spz4* and immune response during mosquito infection. To determine whether Spz4 is involved in the mosquito

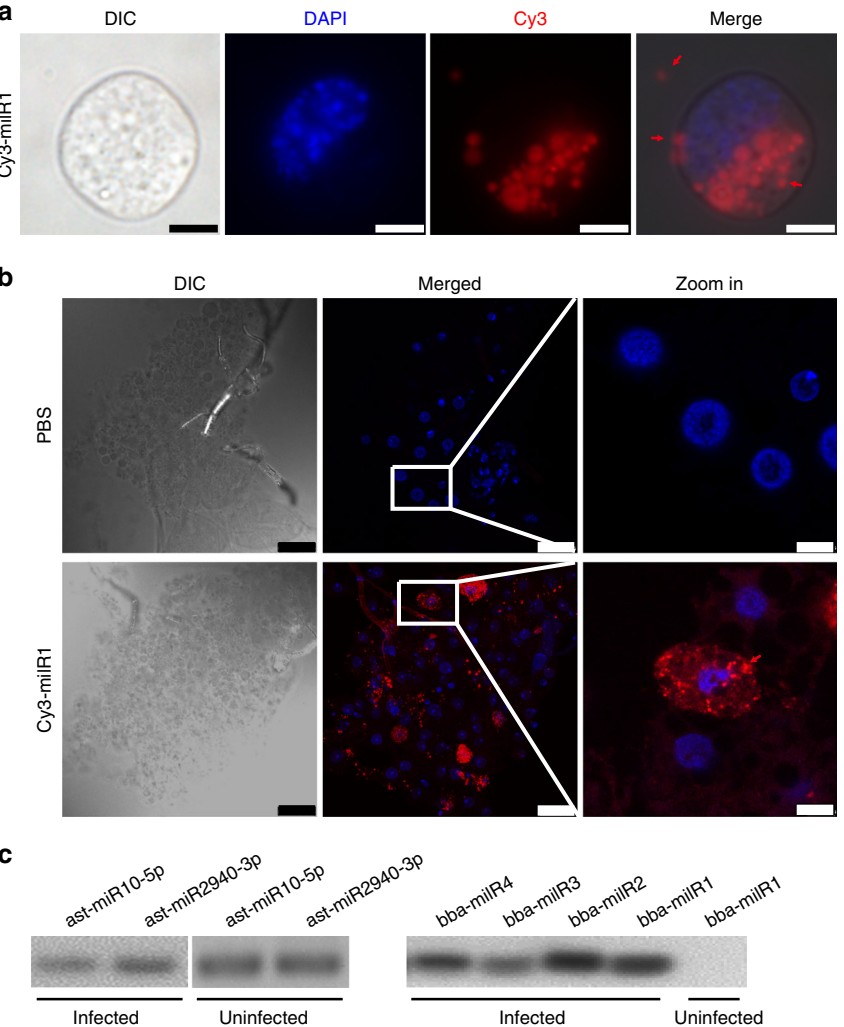

**Fig. 2** bba-milR1 enters host cells and binds to mosquito Argonaute 1 (AGO1). **a** bba-milR1 enters C6/36 mosquito cells via vesicles that were present outside the cells, on the cell surface or in the cytoplasm. Cy3-fluorescein-labeled bba-milR1 (2 μM) was added to C6/36 cells, incubated for 24 h, then washed with PBS, fixed in 4% paraformaldehyde, and stained with DAPI. DIC, differential interference contrast microscopy; red, Cy3; blue, DAPI. Scale bars, 5 μm. **b** bba-milR1 is translocated into adult mosquito fat body cells. Cy3-labeled bba-milR1 (20 μM) or PBS was injected into the hemocoel of *A. stephensi* female mosquitoes. The fat body was dissected 24 h later, washed with PBS, fixed in 4% paraformaldehyde, followed by nucleic acid staining with DAPI. Scale bars in DIC and merge: 25 μm; scale bars in zoom in: 5 μm. Images were acquired by confocal microscopy. **c** RT-PCR detection of AsAgo1-bound sRNAs. Fungal milRNAs are detected in the AsAGO1-RIP (RNA immunoprecipitation) fraction from the *B. bassiana*-infected *A. stephensi* mosquitoes that were collected at 36 hpi, 60 hpi and 84 hpi, mixed and homogenized in ice-cold RIP lysis buffer, but not in the uninfected mosquitoes. AGO1-bound sRNAs were pulled down from fat body homogenates with antibody AsAGO1-linked magnetic beads. RNA was then released from AGO1-RIP fraction after digestion with protease K. Uninfected mosquitoes mixed with *B. bassiana* ARSEF252 mycelia were used as a control to rule out any nonspecific association between AGO1 and milRNAs during the experimental process. The AGO1-associated sRNAs were detected by RT-PCR. As a positive control, two *A. stephensi* miRNAs ast-miR10-5p and ast-miR2940-3p were detected in AsAGO1-RIP samples. Source data are provided as a Source Data file

Toll immune signaling pathway, we silenced *Spz4* by systemic injection of *Spz4* double-stranded RNA (dsSpz4). The mRNA levels of *CEC1* and *DEF1*, which are antimicrobial peptide genes downstream of the Toll pathway, were reduced by ~50% in dsSpz4-treated mosquitoes compared with the dsGFP-treated controls (Fig. 4b). The *CEC1* and *DEF1* transcript levels were decreased by ~70% in agomir-injected mosquitoes compared with negative controls, which provides further evidence for *Spz4* repression and inhibition of the Toll signaling pathway by bba-milR1 (Fig. 4c). To define the contribution of Spz4 in the mosquito immune defense against *B. bassiana* infection, we knocked down *Spz4* by injecting dsSpz4 and then infected mosquitoes with *B. bassiana*. At 60 h after topical infection of *B. bassiana*, the formation of hyphal bodies, a proxy for fungal invasion efficiency,

was significantly more abundant in dsSpz4-treated mosquitoes than the dsGFP control (Fig. 4d). The fungal biomass in dsSpz4-treated mosquitoes was 2.2-fold greater than the dsGFP controls (Fig. 4e). Moreover, silencing of *Spz4* rendered mosquitoes more susceptible to the *B. bassiana* infection than the dsGFP-treated control, and the survival rate was significantly lower (Fig. 4f). These results indicate that Spz4, a Toll receptor ligand, regulates the expression of the antifungal peptide genes to protect mosquitoes from fungal infection. We conclude that bba-milR1 acts as a fungal effector molecule to suppress mosquito immune response by targeting host *Spz4*.

Melanization, another important immune mechanism of arthropods, also plays an important role in the innate immune response to encapsulate and retard invasive *B. bassiana* growth

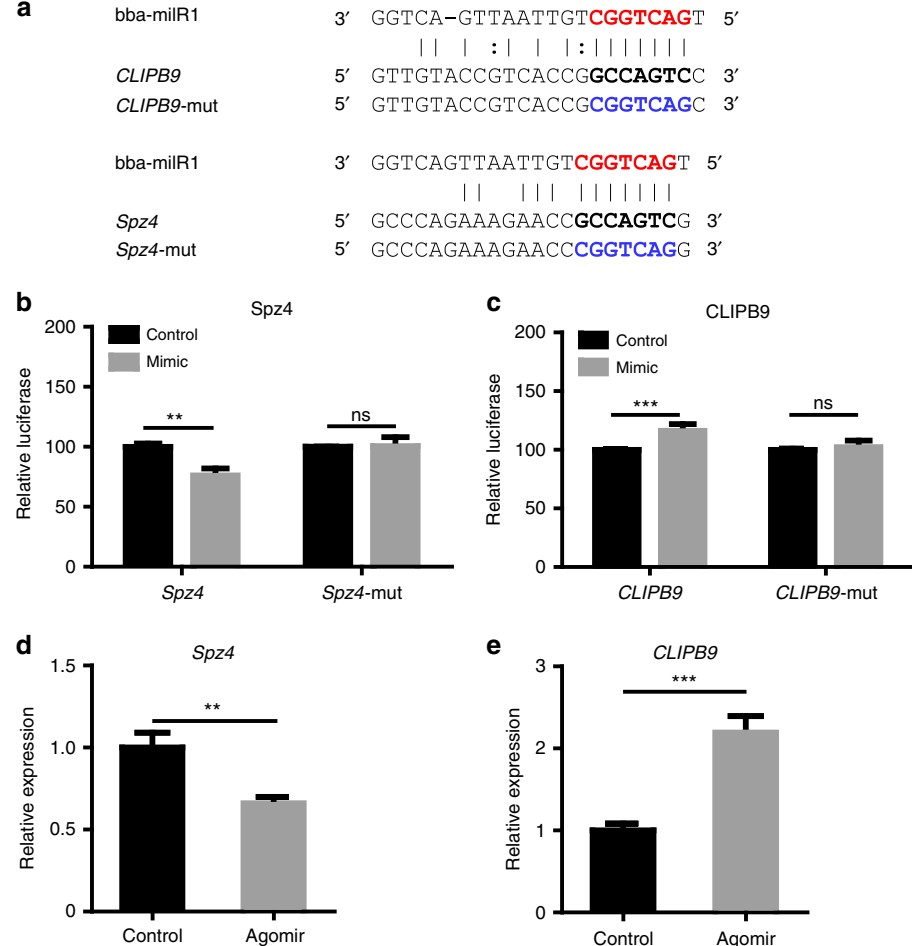

**Fig. 3** bba-milR1 targets the mosquito genes *Spz4* and *CLIPB9*. **a** The sequences of bba-milR1 target sites in the coding regions of *Spz4* and *CLIPB9* genes of *A. stephensi* mosquito. Red characters are seed regions of bba-milR1. Black bold characters are target sequences. The mutated forms of target genes are shown in blue characters. **b** bba-milR1 suppresses Spz4 expression. **c** bba-milR1 induces CLIPB9 expression. The interactions between bba-milR1 and the target sites of *Spz4* and *CLIPB9* in *A. stephensi* were determined by dual-luciferase reporter assays in HEK293T cells that were co-transfected with bba-milR1 mimics and psiCHECK2 vectors containing wild or mutant target sequences of Spz4 or CLIPB9 genes. **d** The transcript level of *Spz4* is suppressed in mosquitoes injected with bba-milR1 agomir (200 nM). The mosquitoes injected with control agomir (200 nM) served as a negative control. Gene expression was quantified by qRT-PCR. **e** The transcript level of *CLIPB9* is upregulated in mosquitoes injected with bba-milR1 agomir compared with the control. Values are mean ± s.e.m. The expression values are normalized to control. The experiments were repeated three times with similar results. **P < 0.01, ***P < 0.001, ns, not significant (Student's *t* test). Source data are provided as a Source Data file

and dissemination in mosquitoes[13]. Melanization is regulated by activation of prophenoloxidase (PPO) into phenoloxidase (PO), and involves an extracellular proteinase cascade and serpin inhibitors (Fig. 5a). The clip domain serine proteinase CLIPB9 acts as a PPO-activating proteinase to regulate melanization in mosquitoes[26]. To confirm whether *CLIPB9* is indeed modulated by bba-milR1 during infection, we examined the transcript levels of *CLIPB9* in mosquitoes after infection with WT, milR1-OV, and milR1-KO strains. The results showed that there was no significant difference in the transcript levels of *CLIPB9* between mosquitoes at 36 h post infection with WT, milR1-KO, and milR1-OV (Fig. 5b; Supplementary Fig. 10a), indicating that bba-milR1 does not interact with *CLIPB9* during the early stages of infection when fungus penetrating the mosquito integument. However, the expression of *CLIPB9* was significantly higher and lower in the mosquitoes infected by *B. bassiana* milR1-OV and the milR1-KO mutant, respectively, than the WT-infected mosquitoes at late stages of infection (84 hpi) (Fig. 5c; Supplementary Fig. 10b). To test the role of CLIPB9 in the defense against *B. bassiana* infection, we silenced *CLIPB9* by

systemic injection of *CLIPB9* dsRNAs. The dsCLIPB9 silencing reduced *CLIPB9* mRNA levels by 83% (Supplementary Fig. 11a). Then, we measured the hemolymph PO activity in dsGFP-treated and dsCLIPB9-treated mosquitoes at 3 days post injection. The PO activity was significantly lower in dsCLIPB9-injected mosquitoes than dsGFP-injected mosquitoes (Supplementary Fig. 11b). Similarly, infection of milR1-OV and milR1-KO resulted in significant increase and decrease in *A. stephensi* hemolymph PO activity compared with WT at 84 hpi, respectively (Fig. 5d). As a consequence of *CLIPB9* knock down, the number of *B. bassiana* hyphal bodies was higher than in dsGFP-treated controls (Fig. 5e). qPCR analysis demonstrated that fungal load was significantly higher in dsCLIPB9-treated mosquitoes compared with the dsGFP-treated controls (Fig. 5f). Moreover, silencing of *CLIPB9* rendered mosquitoes more susceptible to *B. bassiana* infection than dsGFP-treated controls (Fig. 5g). Taken together, these results show that *B. bassiana* markedly reduces bba-milR1 expression during late stages of infection to elaborately avoid induction of *CLIPB9*, and in this way circumvent the melanization response (Fig. 6).

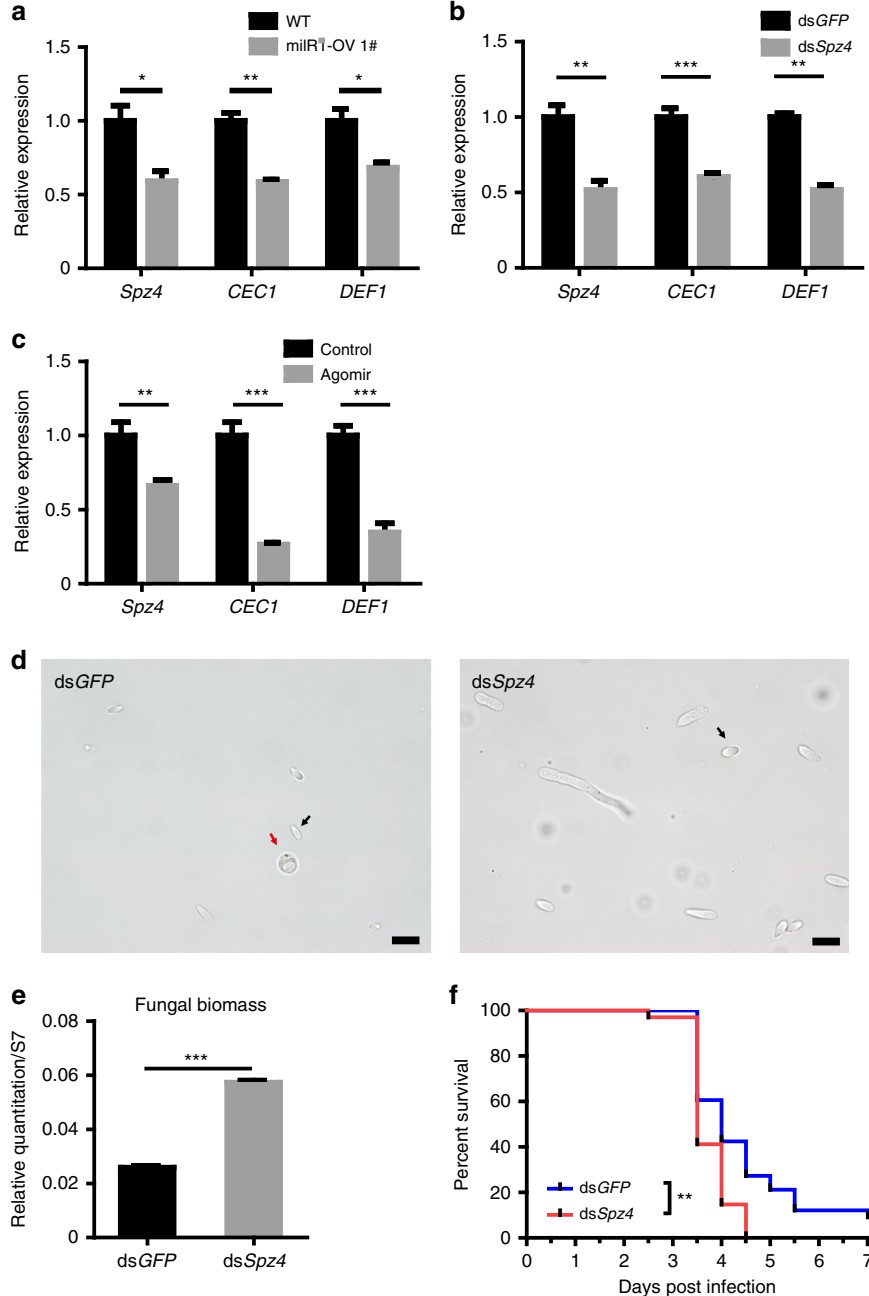

**Fig. 4** bba-milR1 attenuates host immunity by suppressing Toll receptor ligand Spätzle 4. **a** The relative transcript levels of *Spz4*, *cecropin 1* (*CEC1*), and *defensin 1* (*DEF1*) in mosquitoes at 48 h post infection (hpi) with WT and bba-milR1-overexpressing strain milR1-OV 1#. The expression values are normalized to WT. **b** Effect of *Spz4* silencing on the expression of *CEC1* and *DEF1*. The expression values are normalized to dsGFP. **c** The relative transcript levels of *Spz4*, *CEC1*, and *DEF1* in the mosquitoes injected with bba-milR1 agomir (200 nM) or control agomir (200 nM). The expression values are normalized to control. **d** Effect of *Spz4* silencing on fungal hyphal body formation in the mosquito hemocoel at 60 hpi. Black arrows point to hyphal bodies, and the red arrow points to a mosquito hemocyte. Scale bars, 10 μm. **e** qPCR-based quantification of fungal load in mosquitoes injected with dsGFP or dsSpz4 at 60 hpi. Fungal levels are expressed as that of fungal 18S rRNA relative to *A. stephensi* ribosomal protein S7 (AsS7) DNA. **f** Effect of *Spz4* silencing on the survival of mosquitoes following topical application of a suspension of $10^7$ conidia/ml of *B. bassiana* ARSEF252 (Log-rank test). Values are mean ± s.e.m. The experiments were repeated three times with similar results. *$P < 0.05$, **$P < 0.01$, ***$P < 0.001$. *P*-value < 0.05 was regarded as statistically significant (Student's *t* test). Source data are provided as a Source Data file

## Discussion

*Beauveria bassiana* has been at the forefront of efforts to develop biocontrol alternatives to the use of chemical insecticides for vector control[5,10,27,28]. This fungus is also widely used as a model for studies of fungal pathogenicity and fungus–invertebrate host molecular interactions[14,20]. Insects possess a refined innate immune system capable of recognizing fungal pathogens[12,25]. To develop in insects, fungal pathogens must be able to attenuate their host immune responses. How this happens remains elusive. Here, we show that the fungal pathogen produces a sRNA effector that is translocated into host insect cells to attenuate host immunity and achieve its infection, which is an example of the evolution of a cross-kingdom RNA-mediated defense mechanism.

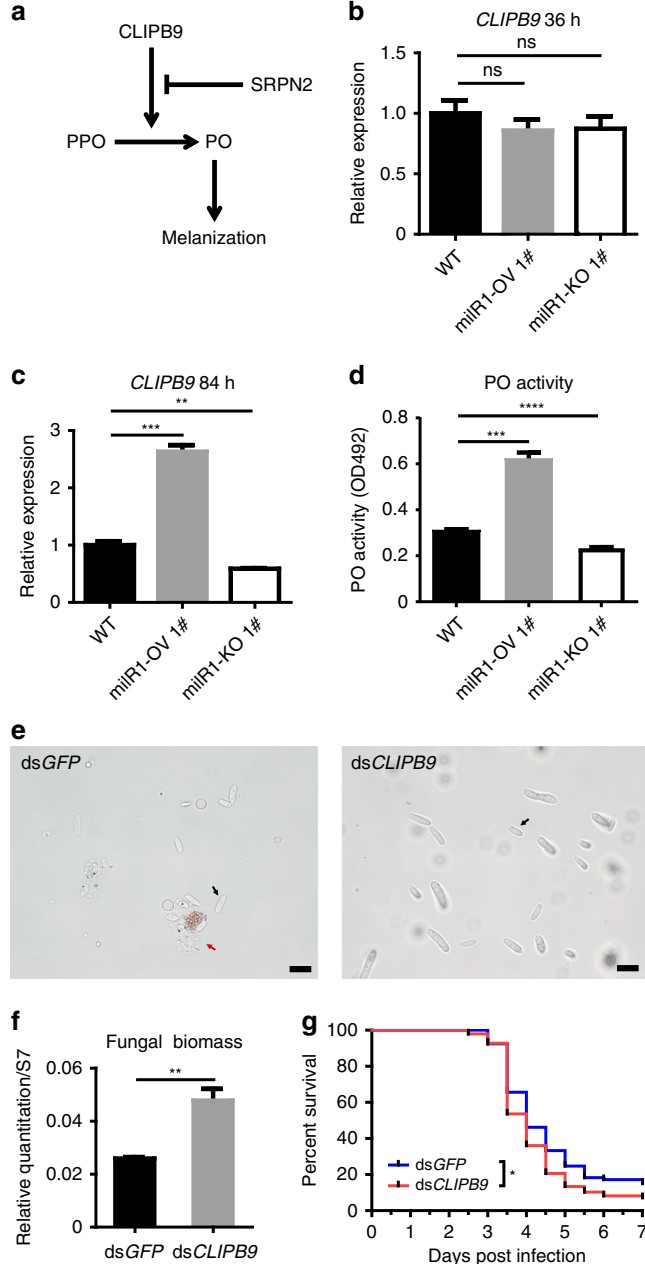

**Fig. 5** The interaction of bba-milR1 with the target gene *CLIPB9*. **a** Diagram of the mosquito melanization response. Clip domain serine proteinase: CLIPB9; Serine protease inhibitor: SRPN2; prophenoloxidase: PPO; phenoloxidase: PO. **b** The relative transcript levels of *CLIPB9* in mosquitoes at 36 h post infection with WT, milR1-OV 1#, or milR1-KO 1# strains. There was no significant difference in *CLIPB9* expression among the groups at this time point (Student's *t* test). **c** The relative transcript levels of *CLIPB9* in mosquitoes at 84 h post infection with WT, milR1-OV 1#, or milR1-KO 1# strains. The expression values are normalized to WT. **d** Infection of milR1-OV1# and milR1-KO 1# results in significant increase and decrease in *A. stephensi* hemolymph PO activity compared with WT at 84 h post infection, respectively. **e** Effect of *CLIPB9* silencing on fungal hyphal body formation in the mosquito hemocoel at 60 hpi. The black arrows and red arrow indicate the hyphal bodies and mosquito hemocyte, respectively. Scale bars, 10 μm. **f** qPCR-based quantification of fungal load in mosquitoes injected with dsGFP or dsCLIPB9 at 60 hpi. Fungal levels are expressed as that of fungal 18S rRNA relative to *A. stephensi* ribosomal protein S7 (AsS7) DNA. **g** Effect of *CLIPB9* silencing on the survival of mosquitoes following topical application of a suspension of $10^7$ conidia/ml of *B. bassiana* ARSEF252 (Log-rank test). Values are mean ± s.e.m. Similar results were obtained in three biological repeats. *$P < 0.05$, **$P < 0.01$, ***$P < 0.001$. *P*-value < 0.05 was regarded as statistically significant (Student's *t* test). Source data are provided as a Source Data file

Small RNAs play critical roles in regulating gene expression in many organisms. In insects, miRNAs associate with AGO1 to mediate gene regulation[24]. Our immunoprecipitation assays demonstrated that the bba-milR1 can enter the insect cells where it associates with mosquito AGO1 and in this way hijacks the insect RNAi machinery to selectively modulate host defense genes. The bba-milR1 was found in exosome-like extracellular vesicles (EVs) from where it is transported into mosquito and fruit fly cells and fat body tissues. Recent studies suggest that EV-mediated export of sRNA represents a conserved universal mechanism for intra-kingdom and inter-kingdom communication[29,30].

Insect pathogenic fungi gain access to the insect hemocoel by penetrating its chitinous integument[20]. *B. bassiana* penetrates the mosquito integument at ~36 hpi, and invades the mosquito's hemocoel at ~60 hpi[31]. To successfully develop in their insect hosts, pathogenic fungi must be able to disable host immune responses. The cleaved ligand Spätzle activates the Toll immune pathway, which is the principal insect antifungal mechanism[12].

In insects, the Toll is activated early by recognizing fungal structures while traversing the external cuticle[32,33]. We found that *A. stephensi* Spz4 is highly expressed in the mosquito integument at the point of fungus penetration. Interestingly, we found that the bba-milR1 is highly induced and expressed in *B. bassiana* during cuticle penetration, so that it can efficiently silence *Spz4* and suppress Toll-mediated mosquito immunity.

Once in the hemocoel, fungal hyphae encounter host cellular and humoral defenses. To counter these defenses, the invasive filamentous cells switch to a yeast-type proliferation strategy to form hyphal bodies that possess fewer carbohydrate epitopes, allowing fungi to avoid recognition by the host immune system[14,20]. The other target of bba-milR1 is *A. stephensi CLIPB9* that regulates PPO activation in the melanization response[26]. Although bba-milR1 can induce expression of *CLIPB9*, we found that *B. bassiana* reduces bba-milR1 expression to very low level after entering the hemocoel, thus elaborately avoiding direct induction of *CLIPB9*. If bba-miR1 is overexpressed under the control of a constitutive promoter, it impairs fungal pathogenicity (Fig. 1c). CLIPB9 and PPO are mostly expressed in mosquito hemocytes[34,35], which could explain why highly expressed bba-milR1 during fungal penetration of the insect integument did not target and induce expression of *CLIPB9*. These findings uncover how the fungal pathogen manipulates its host's defense mechanisms by deploying bba-milR1 in a spatial and temporal manner at different phases of infection (Fig. 6). This study also opens a new avenue to improve fungal pathogenicity against mosquitoes, leading to better control of mosquito-borne diseases.

## Methods

**Mosquito rearing.** *Anopheles stephensi* (Dutch strain) mosquitoes were maintained at 27 °C with 70 ± 5% relative humidity under 12 h/12 h day–night cycle. Larvae were fed on cat food pellets and ground fish food supplement. Adult mosquitoes were maintained on 10% (wt/vol) sucrose.

**Fungal culture and infection bioassay.** *Beauveria bassiana* strains ARSEF252, ARSEF2860, and Bb-bm01 were grown and maintained on Sabouraud dextrose agar plus yeast extract (SDAY; BD Difco) at 26 °C. Conidia were obtained from 10-day-old SDAY cultures. Conidia suspensions were prepared in 0.01% (vol/vol) Triton X-100 and filtered through layers of sterile glass wool to remove hyphal fragments. To conduct fungal infection, 5-day-old female *A. stephensi* adult

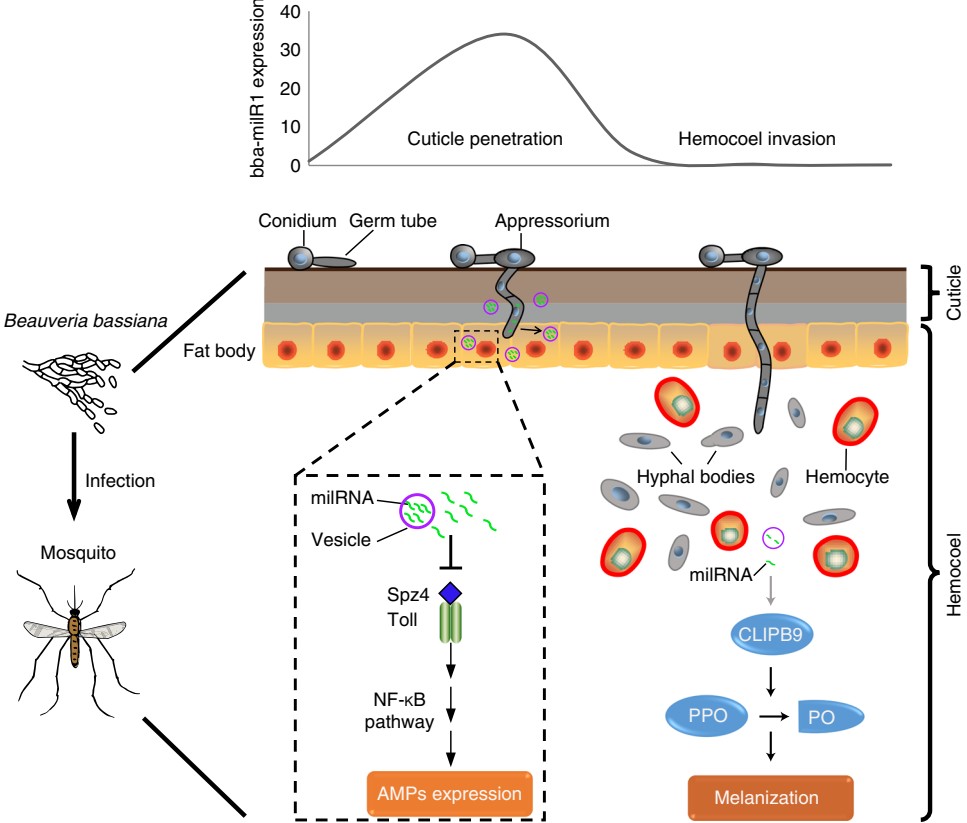

**Fig. 6** Model of *Beauveria bassiana* deploying bba-milR1 to modulate mosquito immunity. During early stages of infection, *B. bassiana* bba-milR1 is highly expressed, and translocated into the mosquito cells to attenuate mosquito immune responses by suppressing the expression of the key activator gene *Spz4*. At this early stage in the integument, the bba-milR1 is not accessible to circulating hemocytes, the site of *CLIPB9* gene expression. During later infection stages when the fungus enters the hemocoel, *B. bassiana* markedly decreases the expression of bba-milR1 to elaborately avoid induction of *CLIPB9* and activation of melanization via conversion of prophenoloxidase (PPO) to phenoloxidase (PO) (see Fig. 5a)

mosquitoes were sprayed with fungal conidial suspension ($10^7$–$10^8$ conidia/mL). Mosquitoes sprayed with sterile 0.01% Triton X-100 were used as a negative control. The treated mosquitoes were subsequently maintained at 26 ± 1 °C and 80 ± 5% relative humidity, with a 12-h/12-h day–night cycle, until mosquitoes died or were collected for sample preparation at the indicated time points. Each treatment was replicated three times with 50 mosquitoes per replicate, and the infection bioassays were repeated three times. Mortality was recorded every 12 h, and cadavers were transferred to moisturized filter paper to monitor the emergence of fungal hyphae.

**Small RNA library construction and sRNAs sequencing**. Mosquito samples were collected at 36 , 60 , and 84 h after topical infection with *B. bassiana* ARSEF252. For each time point, about 50 infected mosquitoes collected from two biological replicates were pooled. The fungus-infected mosquito samples were homogenized with beads in RNAiso Plus (TaKaRa). The total RNA was extracted using Direct-zol RNA Miniprep Kit (Zymo Research Corporation) and treated with DNase I (TaKaRa) following the manufacturer's instructions. The RNA concentration and purity were assessed in an Agilent 2100 Bioanalyzer (Agilent) to verify RNA integrity. Small RNA libraries were constructed using a TruSeq small RNA sample preparation kit (Illumina) following the manufacturer's instructions. Illumina microRNA sequencing was performed at Shanghai Biotechnology Corporation. Briefly, the 3′ and 5′ RNA adapters were ligated to the corresponding ends of small RNAs. Following adapter ligation, the ligated RNA fragments were reverse transcribed using M-MLV reverse transcriptase (Invitrogen). The resulting cDNA products were PCR amplified with two primers that were complementary to the ends of the adapter sequences. The PCR amplicons were separated by size in 6% polyacrylamide gel for sRNA (20–30 nt) enrichment and sequenced on Illumina HiSeq 2500 sequencing system.

**miRNA sequence analysis**. The raw reads from sequencing data were filtered by removing poor quality reads, adaptor pollution reads and reads less than 18 nt. The clean reads of small RNAs were aligned to the reference *Beauveria bassiana* genome. The alignment analysis was conduced using CLC Genomics Workbench 12. The sequences that correspond to known miRNAs were determined by matching to the miRNA database (miRBase 22.0). The unannotated sRNA sequences were

aligned to the reference *B. bassiana* genome to find precursor sequences for novel miRNAs. Novel miRNAs were predicted by miRCat2 (http://srna-workbench.cmp.uea.ac.uk/mircat2/) with stem-loop structure. The R package DEGseq software was used to analyze differentially expressed miRNAs.

**miRNA target prediction**. Three miRNA prediction softwares miRanda (http://www.microrna.org) and PITA (http://genie.weizmann.ac.il/index.html) and microTar (http://tiger.dbs.nus.edu.sg/microtar/) were jointly used to predict miRNA targets in mosquito *Anopheles stephensi*. Thresholds were set to a score of ≥ 140 for miRanda (default), ddG ≤ 0 for PITA, and energy ≥ 0.5 for microTar.

**Fungal small RNAs deletion and overexpression**. For targeted deletion of milRNAs (bba-milR1, bba-milR2, bba-milR3, or bba-milR4), the 5′ and 3′ flanking regions of the miRNAs were amplified by PCR from *B. bassiana* ARSEF252 genomic DNA and then subcloned into the *Xba*I and *Eco*RV sites of the binary vector pBarGFP[36]. The gene disruption vectors were then transformed into *Agrobacterium tumefaciens* AGL-1 for targeted gene disruption in *B. bassiana* ARSEF252 by homologous recombination[16].

To overexpress miRNAs (bba-milR1, bba-milR2, bba-milR3, or bba-milR4) in *B. bassiana* ARSEF252, the ~400 bp fragment surrounding primary milRNA was amplified by PCR from *B. bassiana* genomic DNA, and then subcloned into the *Eco*RI and *Eco*RV sites of the binary vector pBarGFP-PgpdA downstream of a constitutive *Aspergillus nidulans* gpdA promoter (PgpdA), to generate microRNA expression vectors. The sequence of the microRNA amplicon was confirmed by sequencing. The vectors were separately transformed into wild-type *B. bassiana* ARSEF252 using *Agrobacterium tumefaciens*-mediated transformation to generate microRNA overexpression strains[36].

**Generation of Dicer mutants of *B. bassiana***. To generate *Dicer1*, *Dicer2* single knockout mutant, the 5′ and 3′ flanking regions of the genes open-reading frame were amplified by PCR from *B. bassiana* ARSEF252 genomic DNA, and then subcloned into the binary vector pBarGFP. *Dcl1* and *Dcl2* deletion mutants were generated by using homologous recombination and the *Agrobacterium tumefaciens*-mediated transformation system[36]. To double-knockout *Dcl1* and *Dcl2*, the 5′ and 3′ flanking regions of *Dcl2* ORF were subcloned into the binary vector

pSurGFP. The resulting vector was applied to *A. tumefaciens*-mediated transformation in a *Dcl1* mutant strain as described above. The sequences of the primers are given in Supplementary Data 1.

**Mosquito RNA isolation and qPCR**. The total RNA was extracted from mosquitoes by using RNAiso Plus (TaKaRa) and treated with Recombinant DNase I (TaKaRa) according to the manufacturer's instruction. For mRNA, cDNAs were synthesized from the total RNA using a PrimeScript™ RT reagent Kit (TaKaRa). Quantitative reverse transcription PCR (qPCR) reactions were performed using the AceQ qPCR SYBR Green MasterMix Kit (Vazyme). cDNAs for miRNAs were reverse transcribed by using miRcute miRNA First-Strand cDNA Synthesis Kit (Tiangen), and qRT-PCR reactions were performed using miRcute miRNA qPCR detection kit (Tiangen) according to the manufacturer's protocol. Each sample was performed in triplicate. The housekeeping gene RPS7 (AsS7) and snRNA U6 were used as endogenous control for mRNA and miRNA, respectively.

**Luciferase reporter assays**. For high-transfection efficiency and low background expression of bba-milR1, the mammalian HEK293T cell line (ATCC) was used for luciferase reporter assay[37]. The HEK293T cells were grown in the DMEM/HIGH GLUCOSE medium (HyClone) containing 10% (vol/vol) heat-inactivated fetal bovine serum (FBS, Gibco) and 1 × antibiotic–antimycotic (Gibco) at 37 °C under 5% CO$_2$. The ~400 bp sequences surrounding the predicted bba-milR1 target sites in *CLIPB9* and *Spz4* were separately cloned into the *Xho*I and *Not*I sites of the psiCheck-2 vector (Promega). Mutagenesis PCR was performed to introduce point mutations at the bba-milR1 target sites to construct psiCheck-2-mut vectors. The HEK293T cells were transfected with 100 ng of psiCheck-2 reporters with 100 nM (final concentration) of synthetic bba-milR1 Mimic (sense strand 5′-UGA-CUGGCUGUUAAUUGACUGG-3′, anti-sense strand 5′-AGUCAAUUAA-CAGCCAGUCAUU-3′, RiboBio) or Negative Control miRNA Mimic (micrON mimic NC #24, RiboBio) using Attractene Transfection Reagent (Qiagen). Cells were collected and lysed 48 h after transfection. The activities of firefly and Renilla luciferases were measured using the Dual-Luciferase Reporter Assay System (Promega). Each sample was performed in triplicate, and transfections were repeated three times.

**miRNA Agomir injection**. bba-milR1 Agomir is a chemically modified double-strand stable bba-milR1 (sense strand 5′-UGACUGGCUGUUAAUUGACUGG-3′, anti-sense strand 5′-AGUCAAUUAACAGCCAGUCAUU-3′). The 3-day-old female mosquitoes were microinjected using Nanoject II microinfector (Drummond) into the thorax with 69 nl agomir (200 nM). Control mosquitoes were injected with negative control Agomir (200 nM). bba-milR1 Agomir and control Agomir (micrON agomir NC #24) were synthesized by RiboBio (Guangzhou, China). Mosquitoes were allowed to recover for 2–3 days before fungus infection.

**dsRNA-mediated gene silencing in adult mosquitoes**. To produce the double-stranded RNA of the genes Spz4 (dsSpz4) and CLIPB9 (dsCLIPB9), the coding regions of the Spz4 and CLIPB9 genes were separately PCR amplified from *A. stephensi* cDNA with forward and reverse primers containing the T7 promoter sequence at their 5′ ends (5′-TAATACGACTCACTATAGGG-3′) (Supplementary Data 1). The PCR products were purified with Cycle-Pure Kit (OMEGA) and used as the template to synthesize dsRNA in vitro using the MEGAscript RNAi kit (Life Technologies). The dsRNA was further purified using the purification column supplied with the kit, eluted with nuclease-free water, and concentrated to 3 µg/µl using a Microcon YM-100 filter (Millipore). An enhanced green fluorescent protein (eGFP)-derived double-stranded RNA (dsGFP) was synthesized and used as a negative control. For dsRNA microinjection, cold-anesthetized 3-day-old female mosquitoes were injected with 69 nl of dsRNA solution (3 ng/µl) into the hemocoel using Nanoject II microinfector (Drummond). Injected mosquitoes were allowed to recover for 2–3 days before conducting fungal infection. To compare the abundance of hyphal bodies in mosquitoes, the hemolymph was collected. An incision was made in the abdomen between the last two segments with a sterile tweezers, and 10 µl perfusion buffer [60% Schneider's Insect Medium (Sigma), 10% FBS, and 30% citrate buffer (pH 4.5; 98 mM NaOH, 186 mM NaCl, 1.7 mM EDTA, and 41 mM citric acid)][38] was slowly injected into the throax. The diluted hemolymph was collected from the incision by a pipette tip. The hemolymph samples were added to glass slides immediately and observed under a microscope. To quantify the load of *B. bassiana* in the dsRNA-treated mosquitoes, 20 whole mosquitoes were collected for every group and ground in liquid nitrogen. The total DNA was extracted by DNeasy Blood & Tissue Kits (Qiagen) according to the manufacturer's instruction. The fungal housekeeping gene 18S rRNA was used to quantify fungi by qPCR analysis.

**PO activity assay**. To collect mosquitoes hemolymph, 30 mosquitoes were surface-sterilized and dissected at the thorax. The hemolymph was collected into a 1.5 -mL Eppendorf tube containing 200 µl 1 × PBS and 0.2 µl of protease inhibitor solution, centrifuged for 20 min at 4 °C, 10,000 rpm. The hemolymph was recovered in supernatant at the volume about 150 µl, and protein concentration was determined using the Pierce BCA Protein Assay Kit (Thermo scientific). Five micrograms of hemolympmh protein were adjusted to 40 µl in PBS containing protease inhibitors and then added 120 µl of L-DOPA solution (20 mM). The

samples were incubated at 29 °C for 30 min. The optical density at 492 nm was measured for each sample against an L-DOPA control that was incubated with PBS. Each experiment was repeated three times.

**RNA immunoprecipitation**. The fat body of the *B. bassiana* ARSEF252 infected mosquitoes at 36 , 60, and 84 hpi were dissected, mixed, and homogenized in ice-cold RIP lysis buffer[39]. Uninfected mosquitoes mixed with *B. bassiana* ARSEF252 mycelia were used as a control. Magnetic beads were pre-incubated with 5 µg of custom-made antibody against *A. stephensi* AGO1 (GenScript) (Supplementary Fig. 12) or normal mouse IgG (Millipore). Then, the antibody-coupled magnetic beads were incubated with 100 µl mosquito homogenates. The immunoprecipitates were pulled down and digested with protease K to release the bound sRNAs. Finally, the cDNAs were reverse transcribed using miRcute miRNA First-Strand cDNA Synthesis Kit (Tiangen). RT-PCR reactions were performed using a miRcute miRNA qPCR detection kit (Tiangen).

**Western blot**. The total proteins from *A. stehpensi* and *B. bassiana* was extracted by RIPA lysis buffer (Beyotime) containing 1 × protease inhibitor cocktail (Beyotime). Proteins were separated in a sodium dodecyl sulfate-polyacrylamide gel electrophoresis (SDS-PAGE) and transferred onto a PVDF membrane (Bio-Rad). The membrane was blocked with 5% nonfat milk in TBST buffer (pH 7.5; 100 mM Tris·HCl, 150 mM NaCl, and 0.05% vol/vol Tween 20) and incubated overnight 4 °C with AsAgo1 antibody (1:1000, GenScript). The horseradish peroxidase (HRP)-linked goat anti-rabbit IgG secondary antibody (BBI, D110058) was used at a 1:10,000 dilution. Protein bands were detected using SuperSignal West Pico PLUS Chemiluminescent Substrate (Thermo Fisher Scientific).

**Immunohistochemistry to detect miRNA entering host cells**. Mosquito *Aedes albopictus* C6/36 (ATCC) and *Drosophila melanogastor* S2 cell lines (Thermo Fisher Scientific) were used for miRNA transportation assay. The mosquito cells C6/36 cells were grown in the RPMI 1640 Medium (Hyclone) containing 10% (vol/vol) heat-inactivated FBS (Gibco) and 1 × antibiotic–antimycotic (Gibco) at 28 °C. S2 cells were grown in Schneider's Insect Medium (Sigma) containing 10% (vol/vol) heat-inactivated FBS (Gibco) and 1 × antibiotic–antimycotic (Gibco) at 28 °C. To detect the transportation and subcellular localization of exogenous miRNAs in insect cells, C6/36 and S2 cells were cultured in the presence of 2 µM synthesized Cy3-labeled bba-milR1 (GenePharma) for 24 h. After incubation, all cells were terminated by washing with ice-cold PBS and fixed cells with 4% (m/vol) paraformaldehyde (PFA) for 10 min. Then, cells were washed with 1 × PBST for three times and kept in 1 × PBS. For confocal observation, cells were mounted with VECTASHIELD Antifade Mounting Medium containing DAPI (Vector Laboratories, H-1200) for 5 min and fluorescence signals were visualized with Leica SP8 confocal microscope system.

To examine the translocation and localization of bba-milR1 in mosquito for in vivo analysis, 69 nl Cy3-labeled bba-milR1 (20 µM) was injected into female mosquitoes as described above. The mosquito fat bodies were collected in 1 × PBS at 24 h post injection. The tissues were further fixed with PFA, stained with DAPI, and observed with a confocal microscope as described above.

**Statistical analyses**. The statistical significance of the survival data from fungal infection bioassays and Triton X-100 treated mosquitoes (control) was analyzed with a log-rank (Mantel–Cox) test. Other statistical significance was determined by Student's *t* test for unpaired comparisons between two treatments. *P*-value of < 0.05 was regarded as statistically significant. Except for when specified in the context, the results are expressed as mean ± s.e.m. All statistics were performed using GraphPad Prism version 5.00 for Windows (GraphPad Software).

**Reporting summary**. Further information on research design is available in the Nature Research Reporting Summary linked to this article.

## Data availability

The sRNA sequencing datasets have been deposited in the National Center for Biotechnology Information Sequence Read Archive with accession number PRJNA517599. The source data underlying Figs. 1a–d, 2c, 3b–e, 4a–f, and 5b–g, and Supplementary Figs 1c, 2d–j, 3a–c, 4a, b, 5, 7a–e, 8, 9, 10a, b, 11a, b, and 12 are provided as a Source Data file.

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

## Acknowledgements

We thank Prof. Marcelo Jacobs-Lorena at Johns Hopkins University School of Public Health for comments and proofreading the paper. We thank Fang Li for rearing mosquitoes. This work was supported by grants from the Strategic Priority Research Program of Chinese Academy of Sciences (grant XDB11010500), National Key R&D Program of China (2018YFA0900502, 2017YFD0200400), the National Natural Science Foundation of China (grants 31772534, 31830086, 31501703, 3170110113), and One Hundred Talents Program of the Chinese Academy of Sciences (grant 2013OHTP01).

## Author contributions

S.W. conceived the study. S.W. and C.C. designed the experiments. C.C. and Y.W. performed the majority of experiments. Y.W. constructed transgenic and deletion mutants. C.C., Y.W., and P.S. performed bioassays. C.C. and P.S. analyzed and verified target genes. J.Z. performed RNA immunoprecipitation. C.C. and J.L. performed immunohistochemistry. C.C. and P.S. injected dsRNA and observed hyphal bodies. C.C., Y.W., J.L., J.Z., and S.W. analyzed the data. C.C. and S.W. wrote the paper.

## Additional information

**Competing interests:** The authors declare no competing interests.

