## [Peer Review File · Nature Communications]

Reviewers' comments:

Reviewer #1 (Remarks to the Author):

Cui et al identified four miRNA-like small RNAs from the entomopathogenic fungus, *Beauveria bassiana* using small RNA sequencing produced during infection of *Anopheles stephensi* mosquitoes. They showed that production of one of them, bba-miRI-1, was dependent on *dicer2*, but not the others. Knockout or overexpression of bba-miRI-1, but not the other three, affected fungal infection. The authors showed that bba-miRI-1 is highly expressed at 36 hours post-infection that coincides with cuticle penetration during which it targets the transcripts of *Spz4*, an inducer of the Toll pathway, therefore suppressing the expression of antifungal effector molecules. However, later in infection when the fungus gains entrance into the hemocoel, the production levels of bba-miRI-1 is significantly reduced to avoid induction of a clip domain protease (another target of bba-miRI-1) that activates the prophenoloxidase enzyme and melanisation. Therefore, the timing of expression of bba-miRI-1 aligns well with the two different stages of infection. This is a very interesting work and adds to our understanding of the role of small non-coding RNAs in interaction of pathogens and hosts. The investigators have done a thorough study and generally the manuscript is written well, although there are several issues with the usage of the article "the" which is either missing or not needed in many places. In addition, a couple of validations are missing listed below.

- 1) Validation of bba-miRI1 knock out hasn't been shown.
- 2) Fig. 2C: Could authors detect a host miRNA as a control in the uninfected samples to ensure integrity of RNA? I presume the two mosquito control miRNAs shown were detected in the infected samples.

Minor comments:

Line 1: The fungal pathogen should be A fungal pathogen

Line 186 and 213: the statistical significance hasn't been shown in the text or on the figure.

Line 281: ...previously described

Line 367: ... (OMEGA) and used as template to....

Line 379: correct the spelling of fungal

Line 391 and 417: fatbody should be fat body

Page 8 and Fig. 3: mimic and agomir are the same. It might be confusing for readers not familiar with the field when both have been mentioned. I suggest using either mimic or agomir.

Supplementary Table 1: *Spz4*, the target sequences have been shifted and do not align correctly with the miRNA sequences. The word "target" is also misplaced. Further, is there a reason for some residues shown in caps and some in small letters? The lines in between miRNA and target sequences don't line up very well between residues in most instances for the 7 genes.

Sassan Asgari

Reviewer #2 (Remarks to the Author):

Overall this is a well put together paper. However, practical implications of these findings would be strengthened by a phylogenetic analysis of both the evolution of miRNA bba-miRI1 in fungi (there are many genomes available) and the conservation of its target site in Toll-like receptors generally. There would likely not be sufficient selective pressure for a generalist pathogen like *B. bassiana* to evolve a mosquito specific miRNA. This is particularly obvious from the authors results

deriving from a *B. bassiana* isolated from *Bombyx mori*. The target site proposed, and convincingly shown with mutagenesis studies is indeed present in the plant TLR gene Xa21 as shown in this paper (see figure 3 nucleotides 176-182 here: <http://www.plantcell.org/content/plantcell/9/8/1279.full.pdf>). This is particularly relevant as *B. bassiana* is known to colonize plants symbiotically. Taking this broader phylogenetic perspective is not unreasonable and may reveal broader implications for this paper and the miRNAs under investigation.

Reviewer #3 (Remarks to the Author):

The manuscript entitled "The fungal pathogen deploys a small silencing RNA to attenuate mosquito immunity and facilitate infection" by Cui et al. submitted to Nature Communications describes the role of one microRNA-like RNA (miRNA) of the fungus *Beauveria bassiana* in insect infection. The manuscript is well written and it follows a logical development trying to solve the questions that show up through the reading. It provides evidence that the fungus synthesizes miRNAs and the conclusions are attractive. The role of small RNAs in pathogenesis has been described in fungi that infect plants, but this is the first manuscript describing their possible role on the infection of insects. Therefore, the subject of the manuscript is very novel and it would be of interest to a wide research community. In addition, the authors have combined different techniques and the quality of the data and presentation is high. However, the conclusions about the role of a particular miRNA (miR1) in facilitating fungal infection are overstated and more experiments are needed to support them. In addition, the results of some routine experiments should be shown to verify the genotype of some mutants generated in this work. The main and minor issues concerning this manuscript are listed below.

Major concerns

- The identification of miRNAs expressed during mosquito infection. The shown information is scarce. It would be convenient to indicate some details about the types and features of the fungal small RNAs identified in the samples. What sample contained the studied 4 miRNAs? Are these miRNAs expressed by the fungus in axenic cultures?
- Experiments that analyzed mutants (Figure 1, Figure 4 and Figure 5) should have been done with two independent mutants or complemented strains to avoid effects due to ectopic mutations that can occur in transformation or spontaneously.
- One of the most critical results corresponds to figure 1. The difference between miR1-KO and WT strain are very small, although they are statistically significant. This suggests that if miR1 has some effect in infection it is very small and all possible precautions should be considered in any experiment.
- Experiments to demonstrate binding of miR to mosquito Ago need a control experiment. Ago proteins are relatively well conserved, so the authors have to demonstrate that the used antibody is not able to immunoprecipitate *B. bassiana* Ago protein.
- People from plants (Fungal small RNAs suppress plant immunity by hijacking host RNA interference pathways. Weiberg A, Wang M, Lin FM, Zhao H, Zhang Z, Kaloshian I, Huang HD, Jin H. Science. 2013 342:118-23) created an Ago mutant to confirm that the effect of *Botrytis cinerea* small RNAs was mediated by the RNAi mechanism of the plant. They also analyzed the virulence of Dicer mutants of the fungus, which could be done in this manuscript because they are available.
- Explain why the interaction between bba-miR1 and the predicted target genes was assayed in HEK293T instead of insect cells. In addition, the results in figure 3 suggest that human Ago proteins behave in the same way as insect Ago proteins regarding the control of Spz4 and CLIPB9 mRNA levels, which is very surprising, particularly in the case of CLIPB9. Moreover, negative effects of RNAi in expression are well known but positive ones deserve some comments in the manuscript.
- PO activity in figure 5 should include miR1-OV and miR1-KO strains.

- Lines 223-225. "Here, we show that the fungal pathogen transfers a sRNA effector into host insect cells to attenuate host immunity and achieve its infection" This sentence is very blunt and should be rephrased. Insect cells in culture and in the insect are able to uptake naked miR1 but none experiment demonstrates that the fungus secretes miR1 when it is infecting insects.
- Experiments (Southern or PCR) should be added to supplementary material confirming the deletion of dcl1, dcl2, miR1, miR2, miR3 and miR4.
- Consider change the title because it gives the idea that there is only one fungal species that infect insects.

Minor points:

- Many figure legends repeat the conclusion of the experiments without give clues to understand the experiments, requiring surfing in the materials and methods section. Some missed information in legend is stated below, but not all, and the author should look at the legends in detail.
- Figure 1 legend. Part a, indicate the origin of the RNA isolated. Whole mosquito? Fungal mycelium?
- Line 24. Effectively should be removed.
- Line 116. Replace mosquito by *Aedes albopictus*.
- Line 350. Add sequences of bba-miR1 Mimic and Negative Control miRNA Mimic.
- Line 556. Indicate that the whole mosquito was used.
- Line 573. Add the time after infection.
- Line 574. Replace mosquito by fatbody.
- In supplementary material, lines 33-34 should be in part b of the legend figure.

We thank all reviewers for the evaluation of our manuscript and for the constructive comments. We have revised the manuscript taking into account the reviewers' comments. Our point-by-point responses follow. The reviewers' comments are quoted in bold and our responses follow in plain text.

Reviewer # 1 (Remarks to the Author):

1. Cui et al identified four miRNA-like small RNAs from the entomopathogenic fungus, *Beauveria bassiana* using small RNA sequencing produced during infection of *Anopheles stephensi* mosquitoes. They showed that production of one of them, bba-miRI-1, was dependent on *dicer2*, but not the others. Knockout or overexpression of bba-miRI-1, but not the other three, affected fungal infection. The authors showed that bba-miRI-1 is highly expressed at 36 hours post-infection that coincides with cuticle penetration during which it targets the transcripts of *Spzle*, an inducer of the Toll pathway, therefore suppressing the expression of antifungal effector molecules. However, later in infection when the fungus gains entrance into the hemocoel, the production levels of bba-miRI-1 is significantly reduced to avoid induction of a clip domain protease (another target of bba-miRI-1) that activates the prophenoloxidase enzyme and melanisation. Therefore, the timing of expression of bba-miRI-1 aligns well with the two different stages of infection. This is a very interesting work and adds to our understanding of the role of small non-coding RNAs in interaction of pathogens and hosts. The investigators have done a thorough study and generally the manuscript is written well, although there are several issues with the usage of the article “the” which is either missing or not needed in many places.

Response: These comments are much appreciated. We have carefully corrected the inappropriate usage of “the” in the manuscript.

2. In addition, a couple of validations are missing listed below.

1) Validation of bba-milR1 knock out hasn't been shown.

Response: Validation of bba-milR1 knock out and other mutants have been included in Supplementary Figure 2.

2) Fig. 2C: Could authors detect a host miRNA as a control in the uninfected samples to ensure integrity of RNA? I presume the two mosquito control miRNAs shown were detected in the infected samples.

Response: Thanks for the reviewer's insightful suggestion. We have showed that the host miRNAs ast-miR10-5p and ast-2940-3p could be detected in uninfected samples (Fig. 2c).

3. Minor comments:

Line 1: The fungal pathogen should be A fungal pathogen

Response: Thank you for your correction. We have changed "the" to "A".

4. Line 186 and 213: the statistical significance hasn't been shown in the text or on the figure.

Response: The statistical significance has been added in Fig. 4f and Fig. 5g.

5. Line 281: ...previously described

Response: "descripted" has been changed to "described".

6. Line 367: ... (OMEGA) and used as template to....

Response: Thank you for correction. We have changed "(OMEGA), were used as template to" to "(OMEGA) and used as the template to".

7. Line 379: correct the spelling of fungal

Response: "fugal" has been changed to "fungal".

8. Line 391 and 417: fatbody should be fat body

Response: "fatbody" and "fatbodies" in new Lines 394, 420 and Supplementary Fig. 8 have been changed to "fat body" or "fat bodies".

9. Page 8 and Fig. 3: mimic and agomir are the same. It might be confusing for readers not familiar with the field when both have been mentioned. I suggest using either mimic or agomir.

Response: Thank you for your suggestion. In fact, mimic and agomir are similar but not the same. They are double stranded miRNA analogue, but agomir is modified with 2'-O-Me and cholesterol, which has higher stability and activity than mimic. miRNA agomir is more suitable for use in systemic or local injection *In Vivo*. miRNA mimic is usually used in cell transfection. Mimic and agomir have been widely used in insect studies. For example, authors injected miR-276 agomir into *Locusta migratoria* adults (He J, et al.. *MicroRNA-276 promotes egg-hatching synchrony by up-regulating brm in locusts. Proc Natl Acad Sci U S A. 2016 Jan 19;113(3):584-9*). Thereby, we suggest to keep use of mimic and agomir.

10. Supplementary Table 1: Spz4, the target sequences have been shifted

and do not align correctly with the miRNA sequences. The word “target” is also misplaced. Further, is there a reason for some residues shown in caps and some in small letters? The lines in between miRNA and target sequences don’t line up very well between residues in most instances for the 7 genes.

Response: We have re-aligned the miRNA and target sequences in Supplementary Table 1. All nucleotides have been shown in caps.

Reviewer #2 (Remarks to the Author):

Overall this is a well put together paper. However, practical implications of these findings would be strengthened by a phylogenetic analysis of both the evolution of miRNA bba-miR1 in fungi (there are many genomes available) and the conservation of its target site in Toll-like receptors generally. There would likely not be sufficient selective pressure for a generalist pathogen like *B. bassiana* to evolve a mosquito specific miRNA. This is particularly obvious from the authors results deriving from a *B. bassiana* isolated from *Bombyx mori*. The target site proposed, and convincingly shown with mutagenesis studies is indeed present in the plant TLR gene Xa21 as shown in this paper (see figure 3 nucleotides 176-182 here: <http://www.plantcell.org/content/plantcell/9/8/1279.full.pdf>). This is particularly relevant as *B. bassiana* is known to colonize plants symbiotically. Taking this broader phylogenetic perspective is not unreasonable and may reveal broader implications for this paper and the miRNAs under investigation.

Response: We appreciate your positive comments and insightful suggestions. We agree that there would likely not be sufficient selective pressure for *B. bassiana* to evolve a mosquito specific miRNA.

So far, there are few reports on fungal small RNAs. Moreover,

identification of small RNAs based on sequence search may be inaccurate. Similarly, identification of targets of a specific sRNA based on sequence alignment search is also not accurate because we found many predicted targets of bba-milR1 are not the real targets (shown in Supplementary Fig. 7). Although we have analyzed conservation of bba-milR1 target sites in *Spz4* (Toll-like receptor ligand) from various insects. As shown in figure below, there are predicted target sites in *An. gambiae*, *Ae. aegypti*, *D. melanogaster* and *B. mori* *Spz4* genes, but the results need further verification.

So, given the difficulties of identifying other fungal homologies of bba-milR1 and targets of bba-milR1 in other insects, we suggest not to perform a phylogenetic analysis in this study.

```

bba-milR1:      3' GGCAGTTAATTGTCGGTCAGT 5'
                |   ||  |||  |||||
An. stephensi:  5' GCCCAGAAAGAACC GCCAGTCG 3'
bba-milR1:      3' GGCAGTTAATTGTCGGTCAGT 5'
                | |  |  |  |||||
Ae. aegypti:    5' TATTGATAGATTATCCAGTTG 3'
bba-milR1:      3' GGCAGTTAATTGTCGGTCAGT 5'
                || || || |  ||  |||
An. gambiae:    5' CCCGTGAAGGAGGAGGTAGTCG 3'
bba-milR1:      3' GGCAGTTAATTGTCGGTCAGT 5'
                | |                |||||
D. melanogaster:5' CGACCACAGTGCAGCCAGTCC 3'
bba-milR1:      3' GGCAGTTAATTGTCGGTCAGT 5'
                |||  |||||  |
B. mori:         5' AGCACCAAGAGGGAGCCAGACC 3'

```

In-silico analysis of bba-milR1 target sites in *Spz4* from various insects.

Reviewer #3 (Remarks to the Author):

1. The manuscript entitled “The fungal pathogen deploys a small silencing RNA to attenuate mosquito immunity and facilitate infection” by Cui et al. submitted to Nature Communications describes the role of one microRNA-like RNA (miRNA) of the fungus *Beauveria bassiana* in insect infection. The manuscript is well written and it follow a logical development trying to solve the questions that show up through the reading. It provides evidences that the fungus synthesizes miRNAs and the conclusions are attractive. The role of small RNAs in pathogenesis has been described in fungi that infect plants, but this is the first manuscript describing their possible role on the infection of insects.

Therefore, the subject of the manuscript is very novel and it would be of interest of a wide research community. In addition, the authors have combined different techniques and the quality of the data and presentation is high. However, the conclusions about the role of a particular miRNA (miR1) in facilitating fungal infection are overstated and more experiments are needed to support them. In addition, the results of some routine experiments should be shown to verify the genotype of some mutants generated in this work. The main and minor issues concerning this manuscript are listed below.

Response: These comments are much appreciated. We have carefully addressed all your comments (see our responses to your comments listed below).

2. Major concerns

- The Identification of miRs expressed during mosquito infection. The shown information is scarce. It would be convenient to indicate some details about the types and features of the fungal small RNAs identified in the samples. What sample contained the studied 4 miRNAs? Are these miRNAs expressed by the fungus in axenic cultures?

Response: Since we only profiled sRNA libraries generated from fungus-infected mosquitoes collected at 36 h, 60 h and 84 h post fungal topical infection. The sequencing data show that the majority of sRNAs belong to mosquitoes, thereby we only identified 4 miRNA-like small RNAs (miRNAs) whose sequences can be perfectly matched the *B. bassiana* genome (Supplementary Fig. 1a). We have added the length of the sRNAs and their sequencing results in Supplementary Figure 1a. The expression patterns of bba-miR1 during fungus infecting mosquitoes were also shown in Fig 1d. We also showed the predicted secondary structures of these sRNAs in Supplementary Figure 1b. We have shown that these miRNAs are expressed at very low levels by the fungus in axenic cultures, as shown in Supplementary Fig 4a (fungal hyphae) and in Fig 1d "0h" (fungus spores).

3. - Experiments that analyzed mutants (Figure 1, Figure 4 and Figure 5) should have been done with two independent mutants or complemented strains to avoid effects due to ectopic mutations that can occur in transformation or spontaneously.

Response: As per the referee's suggestion, we have conducted additional experiments to test the other independent knock-out strain and overexpression strain. All the phenotypes produced by the two independent knock-out and overexpression strains are similar. The virulence results were shown in Fig 1b and 1c. The effects of the other independent knock-out strain and overexpression strain on the expression of the target genes and AMPs were shown in Supplementary Figure 9 and Supplementary Figures 10.

4. - One of the most critical result corresponds to figure 1. The difference between milR1-KO and WT strain are very small, although they are statically significant. This suggests that if milR1 has some effect in infection it is very small and all possible precautions should be considered in any experiment.

Response: We appreciate your comments. In the previous bioassays, we used very high concentration of fungal conidial suspension (10^8 conidia/mL), which resulted in small difference between milR1-KO and WT. We have conducted additional bioassays using lower concentration of fungal conidial suspension (10^7 conidia/mL), and it turns out that there is very big difference in the virulence between milR1-KO and WT (Fig 1b).

5. - Experiments to demonstrate binding of miR to mosquito Ago need a control experiment. Ago proteins are relatively well conserved, so the authors have to demonstrate that the used antibody is not able to immunoprecipitate *B. bassiana* Ago protein.

Response: Thanks for your suggestion. We have used *B. bassiana* proteins as a control experiment. The western blot results have showed that *A. stephensi* Ago1 antibody only recognizes a ~110 kD bands in *A. stephensi* proteins, but

could not recognize the *B. bassiana* Ago1 protein (Fig S12).

6. - People from plants (Fungal small RNAs suppress plant immunity by hijacking host RNA interference pathways. Weiberg A, Wang M, Lin FM, Zhao H, Zhang Z, Kaloshian I, Huang HD, Jin H. Science. 2013 342:118-23) created an Ago mutant to confirm that the effect of *Botrytis cinerea* small RNAs was mediated by the RNAi mechanism of the plant. They also analyzed the virulence of *Dicer* mutants of the fungus, which could be done in this manuscript because there are available.

Response: We have conducted bioassays by infected *Ago1* knock-down (*dsAgo1*) mosquitoes with *B. bassiana* WT strain. The results showed that *dsAgo1* mosquitoes died faster than did *dsGFP*-injected mosquitoes (see figure below). The possible explanation is that the *Ago1* silencing might influence the function of mosquito miRNAs. Previous studies showed that some insect miRNAs positively regulate innate immune responses (*Mazhar Hussain, Sassan Asgari. MicroRNAs as mediators of insect host-pathogen interactions and immunity. Journal of Insect Physiology 70 (2014) 151–158*). For instance, *Aedes albopictus* *ae-miR-2940* upregulate the metalloprotease *m41 FtsH* gene during virus infection (*Slonchak A , Hussain M , Torres S , et al. Expression of Mosquito MicroRNA Aae-miR-2940-5p Is Downregulated in Response to West Nile Virus Infection To Restrict Viral Replication[J]. Journal of Virology, 2014, 88(15):8457-8467*). Thereby, insect Ago mutant is not appropriate to confirm that the effect of *B. bassiana* small RNAs are mediated by the RNAi mechanism of the insect.

Effect of *Ago1* silencing on the survival of mosquitoes following *B. bassiana*

infection. *A. stephensi* Ago1 was depleted by systemic injection of *Ago1* dsRNAs or dsGFP into mosquito hemocoel, and the treated mosquitoes were infected by topical application of a suspension of 10^7 conidia/ml of *B. bassiana* ARSEF252. **** $P < 0.0001$ (Log-rank test).

We have also showed that *B. bassiana* *Dicer* mutants $\Delta Dcl1/\Delta Dcl2$ resulted in decrease in fungal virulence against *A. stephensi* compared to WT (see figure below). One possible explanation could be that $\Delta Dcl1/\Delta Dcl2$ mutants don't generate *bba-milR1* (as shown in Fig 1a in manuscript), thereby $\Delta Dcl1/\Delta Dcl2$ could not repress mosquito *Spz4* expression during infection. However, we could not rule out another possibility that knock-out of *Dicers* affect other unknown miRNAs that might positively or negatively modulate fungal virulence. Based on the above considerations, it may not be suitable to evaluate a specific miRNA's function by analyzing the virulence of *Dicer* mutants of the fungus against insects.

Effect of Dicers on fungal virulence. Survival of adult female *A. stephensi* mosquitoes infected with the wild-type (WT) ARSEF252 and two $\Delta Dcl1/\Delta Dcl2$ mutant strains following topical application of a spore suspension (10^7 conidia/ml). Mosquitoes sprayed with sterile 0.01% Triton X-100 were used as a negative control (Triton).

7. Explain why the interaction between *bba-milR1* and the predicted target genes was assayed in HEK293T instead of insect cells. In addition, the results in figure 3 suggest that human Ago proteins behave in the same way as insect Ago proteins regarding the control of *Spz4* and *CLIPB9* mRNA levels, which is very surprising, particularly in the case of *CLIPB9*. Moreover, negative effects of RNAi in expression are well known

but positive ones deserve some comments in the manuscript.

Response: In many previous studies of insect miRNAs, researchers performed dual-luciferase reporter assays to verify insect targets of insect miRNA in HEK293T cells because of high transfection efficiency and no background expression of insect target genes and insect miRNAs in HEK293T cells. For example,

(1) Jiang J, et al. *MicroRNA-281 regulates the expression of ecdysone receptor (EcR) isoform B in the silkworm, Bombyx mori*. *Insect Biochem Mol Biol*. 2013 Aug;43(8):692-700.

(2) Ling L, et al. *MIR-2 family targets awd and fng to regulate wing morphogenesis in Bombyx mori*. *RNA Biol*. 2015;12(7):742-8.

To verify the mosquito targets of fungal bba-milR1, we also used the mammalian HEK293T cell line for luciferase reporter assay.

miRNAs usually suppress the target genes by degrading mRNA or repressing translation. However, recent studies have suggested that some miRNAs can up-regulate target mRNA or activate mRNA translation. For instance, human miR-369-3 directs association of AGO-FXR1 complex with the AREs to activate translation (Vasudevan S, Tong Y, Steitz JA. *Switching from repression to activation: microRNAs can up-regulate translation*. *Science*. 2007, 21;318(5858):1931-4.). In *Aedes aegypti* mosquitoes, aae-miR-375 upregulates the transcript level of the Toll immune pathway component *Cactus* (Hussain M, Walker T, O'Neill SL, Asgari S. *Blood meal induced microRNA regulates development and immune associated genes in the Dengue mosquito vector, Aedes aegypti*. *Insect Biochem Mol Biol*. 2013, 43(2):146-52.) Another study demonstrated that endosymbiont *Wolbachia* induces a host microRNA, aae-miR-2940, to enhance transcript levels and/or the stability of the mRNA of metalloprotease in *A. aegypti* (Hussain M, Frentiu FD, Moreira LA, O'Neill SL, Asgari S. *Wolbachia uses host microRNAs to manipulate host gene expression and facilitate colonization of the dengue vector Aedes aegypti*. *Proc Natl Acad Sci U S A*. 2011, 31;108(22):9250-5.). Our studies also showed that bba-milR1 activates the expression of *CLIPB9*, but the exact mechanism is unknown.

8. - PO activity in figure 5 should include milR1-OV and milR1-KO strains.

Response: Thanks for your suggestion. We have tested *A. stephensi* PO activity after infection of *B. bassiana* WT, milR1-OV and milR1-KO. The results have been included in Fig 5d, showing that milR1-OV infection increased PO activity compared to WT, but milR1-KO repressed PO activity.

9. - Lines 223-225. “Here, we show that the fungal pathogen transfers a sRNA effector into host insect cells to attenuate host immunity and achieve its infection” This sentence is very blunt and should be rephrased. Insect cells in culture and in the insect are able to uptake naked milR1 but none experiment demonstrates that the fungus secretes milR1 when it is infecting insects.

Response: We have rephrased the sentence in new Line 226 to “Here, we show that the fungal pathogen produces a sRNA effector that is translocated into host insect cells to attenuate host immunity and achieve its infection”. In RIP assay, *A. stephensi* Ago1 antibody specifically pulled down bba-milR1 during infection progress, but not in samples mixed *A. stephensi* and *B. bassiana*, which shows that the fungus produces bba-milR1 when it is infecting insects.

10. - Experiments (Southern or PCR) should be added to supplementary material confirming the deletion of dcl1, dcl2, milR1, milR2, milR3 and milR4.

Response: Thanks for your suggestions. We have added the results confirming the deletion of dcl1, dcl2, milR1, milR2, milR3 and milR4 in Supplementary Figures 2d-j.

11. - Consider change the title because it gives the idea that there is only one fungal species that infect insects.

Response: We have changed “the” to “A”.

12. Minor points:

- Many figure legends repeat the conclusion of the experiments without give clues to understand the experiments, requiring surfing in the materials and methods section. Some missed information in legend is stated below, but not all, and the author should look at the legends in detail.
- Figure 1 legend. Part a, indicate the origin of the RNA isolated. Whole mosquito? Fungal mycelium?

Response: Thanks for your suggestions. We have added the information to figure legends. For example, we added “RNA was extracted from fungal mycelium” to figure 1a legend.

13. - Line 24. Effectively should be removed.

Response: We have removed “effectively”.

14. - Line 116. Replace mosquito by *Aedes albopictus*.

Response: We have replaced mosquito by *Aedes albopictus*.

15. - Line 350. Add sequences of bba-miR1 Mimic and Negative Control miRNA Mimic.

Response: We have added bba-miR1 Mimic sequences and Product ID of Negative Control miRNA Mimic in new Lines 351-353.

16. - Line 556. Indicate that the whole mosquito was used.

Response: We have changed the Figure 1d legend to “Expression of bba-miR1 during *B. bassiana* ARSEF252 infecting *A. stephensi*. RNA was extracted from fungus-infected mosquitoes.”

17. - Line 573. Add the time after infection.

Response: We have added “mosquitoes that were collected at 36hpi, 60hpi and 84hpi, mixed and homogenized in ice-cold RIP lysis buffer” in new Lines 581-582.

18. - Line 574. Replace mosquito by fatbody.

Response: We have replaced “mosquito” by “fat body”.

19. - In supplementary material, lines 33-34 should be in part b of the legend figure.

Response: We appreciate all your insightful suggestions. We have moved the lines to Supplementary Figure 2b legend.

REVIEWERS' COMMENTS:

Reviewer #1 (Remarks to the Author):

The authors have addressed my comments.

Reviewer #3 (Remarks to the Author):

The revised manuscript "The fungal pathogen deploys a small silencing RNA to attenuate mosquito immunity and facilitate infection" by Cui et al. has been improved according to the reviewers' criticisms. Moreover, I acknowledge that the authors have provided appropriate responses to all the issues raised by reviewers. However, there are some minor points that should be addressed.

- Indicate references supporting the use of HEK293T cells.
- In Supplementary Figure 2. It is not clear what primers were used in part f. Please, make it clear and indicate size of the DNA marker bands. I recommend using external primers to the gene construction used in transformation in future analyses because they tell you if the fragments have been integration in the right locus.
- Add a more descriptive legend in Supplementary Figure 12. This figure is far from clear as it is now.
- Figure 1d legend. Were the expression values at different times normalized against time 0? If so, indicate it in the legend. The same applies for all of the figures with the same normalization.

We thank all reviewers for the evaluation of our manuscript and for the constructive comments. We have revised the manuscript taking into account all the reviewer's comments. Our point-by-point responses follow. The reviewers' comments are quoted in bold and our responses follow in plain text.

Reviewer # 1 (Remarks to the Author):

1. The authors have addressed my comments.

Response: We appreciate your evaluation.

Reviewer #3 (Remarks to the Author):

1. The revised manuscript “The fungal pathogen deploys a small silencing RNA to attenuate mosquito immunity and facilitate infection” by Cui et al. has been improved according to the reviewers' criticisms. Moreover, I acknowledge that the authors have provided appropriate responses to all the issues raised by reviewers. However, there are some minor points that should be addressed.

Response: We appreciate your evaluation.

2. Indicate references supporting the use of HEK293T cells.

Response: The reference has been cited in Line 348.

3. In Supplementary Figure 2. It is not clear what primers were used in part f. Please, make it clear and indicate size of the DNA marker bands. I recommend using external primers to the gene construction used in transformation in future analyses because they tell you if the fragments have been integration in the right locus.

Response: $\Delta Dcl1/\Delta Dcl2$ double mutants were generated by disrupting *Dcl2* in a *Dcl1* mutant strain, and verified by PCR detecting *Dcl2* gene in the putative

transformants using the two relevant primer pairs of p1 and p2, p3 and p4 (see right diagram of the panel a). We have added the above description in the legend of Supplementary Figure 2. We also added the size of the DNA marker bands in the figure. The sequences of all primers were listed in Supplementary Data 1.

All mutants were verified by two primer pairs. As shown in Supplementary Figure 2a, the relevant p1 and p2 were used to amplify WT genome DNA fragment, p3 and p4 were used to verify knock-out mutants. The WT DNA fragments were not detected in all the mutants, demonstrating that the relevant genes were disrupted. However, we appreciate your suggestion by using external primers in the future analyses.

4. Add a more descriptive legend in Supplementary Figure 12. This figure is far from clear as it is now.

Response: We have changed the figure legend of “AsAGO1 antibody recognizes ~110kD band in the protein sample of *Anopheles stephensi*, but not with the preimmune serum and *B. bassiana* proteins. Source data are provided as a Source Data file” to “The protein sample of *Anopheles stephensi* and *Beauveria bassiana* were applied to western-blot analysis. AsAGO1 antibody recognizes ~110kD band in the protein sample of *A. stephensi*, but not with *B. bassiana* proteins. The preimmune serum of rabbit was used as the negative control. Source data are provided as a Source Data file.”.

5. Figure 1d legend. Were the expression values at different times normalized against time 0? If so, indicate it in the legend. The same applies for all of the figures with the same normalization.

Response: Thank you for your suggestion. In Figure 1d, the expression values at different times were normalized against time 0. We have indicated this in the legends of Figure 1d, Figure 3, 4, 5 and Supplementary Figure 4, 5, 7, 9, 10, 11.